# Prediction of Emergency Cesarean Section Using Machine Learning Methods: Development and External Validation of a Nationwide Multicenter Dataset in Republic of Korea

**DOI:** 10.3390/life12040604

**Published:** 2022-04-18

**Authors:** Jeong Ha Wie, Se Jin Lee, Sae Kyung Choi, Yun Sung Jo, Han Sung Hwang, Mi Hye Park, Yeon Hee Kim, Jae Eun Shin, Ki Cheol Kil, Su Mi Kim, Bong Suk Choi, Hanul Hong, Hyun-Joo Seol, Hye-Sung Won, Hyun Sun Ko, Sunghun Na

**Affiliations:** 1Department of Obstetrics and Gynecology, Eunpyeong St. Mary’s Hospital, College of Medicine, The Catholic University of Korea, Seoul 03312, Korea; wiejh@catholic.ac.kr; 2Department of Obstetrics and Gynecology, Kangwon National University Hospital, Kangwon National University School of Medicine, Chuncheon 24289, Korea; 23wls@naver.com; 3Department of Obstetrics and Gynecology, Incheon St. Mary’s Hospital, College of Medicine, The Catholic University of Korea, Seoul 21431, Korea; obgysk@catholic.ac.kr; 4Department of Obstetrics and Gynecology, St. Vincent’s Hospital, College of Medicine, The Catholic University of Korea, Seoul 16247, Korea; eggs76@catholic.ac.kr; 5Department of Obstetrics and Gynecology, Research Institute of Medical Science, Konkuk University School of Medicine, Seoul 05030, Korea; 20080251@kuh.ac.kr; 6Department of Obstetrics and Gynecology, Ewha Medical Center, Ewha Medical Institute, Ewha Womans University College of Medicine, Seoul 07804, Korea; ewhapmh@ewha.ac.kr; 7Department of Obstetrics and Gynecology, Uijeongbu St. Mary’s Hospital, College of Medicine, The Catholic University of Korea, Seoul 11765, Korea; yoni@catholic.ac.kr; 8Department of Obstetrics and Gynecology, Bucheon St. Mary’s Hospital, College of Medicine, The Catholic University of Korea, Seoul 14647, Korea; jennie1008@catholic.ac.kr; 9Department of Obstetrics and Gynecology, Yeouido St. Mary’s Hospital, College of Medicine, The Catholic University of Korea, Seoul 07345, Korea; kilssine@catholic.ac.kr; 10Department of Obstetrics and Gynecology, Daejeon St. Mary’s Hospital, College of Medicine, The Catholic University of Korea, Seoul 34943, Korea; alex4yu@catholic.ac.kr; 11Innerwave Co., Ltd., Seoul 08510, Korea; bschoi@innerwave.co.kr (B.S.C.); sky_h@innerwave.co.kr (H.H.); 12Department of Obstetrics and Gynecology, School of Medicine, Kyung Hee University, Seoul 05278, Korea; seolhj@khu.ac.kr; 13Department of Obstetrics and Gynecology, Asan Medical Center, University of Ulsan College of Medicine, Seoul 05505, Korea; hswon@amc.seoul.kr; 14Department of Obstetrics and Gynecology, Seoul St. Mary’s Hospital, College of Medicine, The Catholic University of Korea, Seoul 06591, Korea

**Keywords:** emergency, cesarean, nulliparous, prediction, labor, machine learning

## Abstract

This study was a multicenter retrospective cohort study of term nulliparous women who underwent labor, and was conducted to develop an automated machine learning model for prediction of emergent cesarean section (CS) before onset of labor. Nine machine learning methods of logistic regression, random forest, Support Vector Machine (SVM), gradient boosting, extreme gradient boosting (XGBoost), light gradient boosting machine (LGBM), k-nearest neighbors (KNN), Voting, and Stacking were applied and compared for prediction of emergent CS during active labor. External validation was performed using a nationwide multicenter dataset for Korean fetal growth. A total of 6549 term nulliparous women was included in the analysis, and the emergent CS rate was 16.1%. The C-statistics values for KNN, Voting, XGBoost, Stacking, gradient boosting, random forest, LGBM, logistic regression, and SVM were 0.6, 0.69, 0.64, 0.59, 0.66, 0.68, 0.68, 0.7, and 0.69, respectively. The logistic regression model showed the best predictive performance with an accuracy of 0.78. The machine learning model identified nine significant variables of maternal age, height, weight at pre-pregnancy, pregnancy-associated hypertension, gestational age, and fetal sonographic findings. The C-statistic value for the logistic regression machine learning model in the external validation set (1391 term nulliparous women) was 0.69, with an overall accuracy of 0.68, a specificity of 0.83, and a sensitivity of 0.41. Machine learning algorithms with clinical and sonographic parameters at near term could be useful tools to predict individual risk of emergent CS during active labor in nulliparous women.

## 1. Introduction

The global cesarean rate was as high as 21.1% in 2018 and has been continuously increasing from 6.7% in 1990 [1]. In Republic of Korea, as the proportion of elderly gravida and nulliparous women increased, the cesarean section (CS) rate increased from 36.7% in 2012 to 47.3% in 2018 (Korea National Statistical Office Korean Vital Statistics Birth Certificate Data and Vital Statistics Fetal Death File, the Korean Statistical Information Service data (January 2012–December 2018) [2]). However, the increased rate of CS is not associated with a reduction in serious neonatal morbidity, such as cerebral palsy [3,4].

Moreover, emergent CS during labor has shown increased risk of severe maternal and neonatal adverse outcomes compared with elective CS [5,6]. Category 1 CS is defined as urgent CS that takes no longer than 30 min from decision to delivery, in which there is immediate life threat to fetus and mother and an increased likelihood of adverse perinatal complication such as low Apgar score, seizure, neonatal resuscitation, and neonatal intensive care unit admission [7,8]. Moreover, emergent CS in the second stage not only increases maternal and neonatal morbidity, but also increases the risk of preterm birth in subsequent pregnancies [9,10]. Predicting women at high risk for emergent CS is meritorious in obstetric management to reduce maternal and neonatal morbidity.

Many factors have been suggested as risk factors for labor failure in nulliparous women, and several models have been developed to predict the mode of delivery in nulliparous women [11,12,13]. However, these models were developed primarily using data from Western women, and few Asian women were included [11,12,13,14]. Recent machine learning models comprising several algorithms have been shown to provide better prediction of obstetric outcomes including shoulder dystocia and vaginal birth after cesarean delivery [15,16,17].

Some women desire an elective CS during the third trimester due to reasons including fear of childbirth, depression or anxiety about labor and vaginal delivery, and perceptions of having a narrow pelvis or other hereditary factors, rather than objective parameters [18,19]. Several interventions with moderate- or high-certainty evidence mainly targeting healthcare professionals have been useful in reducing CS rates [20]. However, evidence for predicting individual risk of emergent CS due to failure to progress or non-reassuring fetal heart rate (FHR) is very limited, especially during the third trimester and in Asian women [20].

In the current study, our objective was to assess prospectively the use of maternal and fetal, clinical, and ultrasound features to develop an automated machine learning model for unplanned CS in Asian nulliparous women during the third trimester, before the onset of labor.

## 2. Materials and Methods

### 2.1. Study Design

This study was a retrospective cohort study conducted in singleton nulliparous women who delivered between January 2009 and December 2019 at seven hospitals under the College of Medicine, at The Catholic University of Korea. As part of routine obstetric care, obstetricians collect clinical data in electronic medical records (EMR). Data on maternal demographic characteristics and delivery outcomes were collected from the institution’s database via EMR. Data were confirmed, and missing data were abstracted from the chart review by two obstetricians (J.H.W. and H.S.K.).

The study included consecutive nulliparous singleton pregnancies at term (≥37 0/7 weeks of gestation) with cephalic presentation who underwent trial of labor. Exclusion criteria were CS due to non-cephalic presentation, abnormal placentation, disproportionate cephalopelvic size, maternal or fetal problems, elective CS on maternal request, and CS prior to labor onset or active phase (less than 4 cm dilatation of the cervical os with uterine contraction). Patients with maternal complications of overt diabetes, moderate-to-severe pregnancy-associated hypertension (PAH), or fetal congenital anomalies were excluded. Patients with hyper- or hypothyroidism, gestational diabetes (GDM), mild hypertensive disease, mild PAH, or idiopathic thrombocytopenic purpura were not excluded.

### 2.2. Study Population

Clinical and sonographic data of 31,929 women were collected in a retrospective study cohort from the seven hospitals under the College of Medicine. Among them, 25,335 women were excluded due to history of previous delivery, multiple pregnancy, preterm delivery, elective or indicated cesarean delivery, cesarean delivery without definite indication, cesarean delivery before active labor, or missing data. Finally, a total of 6549 term nulliparous women was analyzed (Figure 1).

### 2.3. Variables

Gestational age was calculated from the first day of the last menstrual period and corrected with ultrasonographic information on crown rump length during the first trimester. Ultrasonographic findings were based on biometric findings from 34 weeks 0 days to 39 weeks 6 days’ gestation. Fetal biometry data were biparietal diameter (BPD), abdominal circumference (AC), femur length (FL) [21], and estimated body weight (EBW) calculated using the Hadlock-3 formula.

Emergency CS during labor was noted when emergent CS was performed due to failure to progress or non-reassuring FHR during the active phase of labor. To develop a predictive model of emergent CS during labor, we preselected risk factors based on data obtained from published predictive models [11,13,14]. The preselected risk factors were maternal age, newborn height and weight at delivery, and fetal AC at ultrasonography. We added other variables of maternal weight before pregnancy; fetal BPD, EBW, and FL; gestational age at ultrasonographic examination and at delivery; and presence of GDM or PAH.

### 2.4. Machine Learning

#### 2.4.1. Dataset

All anonymized data were prepared as a learning dataset for the machine learning models and were randomly divided into two sub-cohorts at a 70:30 ratio: (1) a model development cohort (training set) with 4584 observations and (2) an internal validation cohort (validation set) with 1965 observations.

#### 2.4.2. Data Sampling Algorithms

In order to treat significant class imbalance in the data, several sampling strategies were applied to the dataset [22].

##### Random Under-Sampling/Oversampling

This resampling technique for dealing with highly unbalanced datasets consists of removing samples from the majority class (under-sampling) and/or adding more examples from the minority class (oversampling) [23,24].

##### Tomek Link

Tomek links are pairs of nearest neighbors but of opposite classes. Removing the in-stances of the majority class of each pair increases the space between the two classes, facilitating the classification process.

##### Condensed Nearest Neighbor (CNN)

CNN uses sub-data consisting of whole minor class samples and one random sample from the major class. For each of the other samples, the algorithm reclassifies a sample as minor if it is closer to the minor samples than to the random major sample.

##### One-Sided Selection (OSS)

OSS removes noise samples in the data with Tomek-link-based under-sampling, applying CNN to the major group at the same time.

##### Edited Nearest Neighbors (ENN)

OSS removes noise ENN pushes the threshold that classifies the label of the sample to the overrepresented side, supporting the asymptotic performance of a nearest neighbor rule [25].

##### Neighbor Cleaning Rule (NCR)

The NCR algorithm modifies the risk of ENN by combining it with CNN. The algorithm conducts two-sided cleaning, as it decreases the over-presented side samples and increases the samples of minor group.

##### Synthetic Minority Oversampling Technique (SMOTE)

The NCR algorithm SMOTE selects two minority class examples from the feature space and creates a synthetic sample that connects the two selected example sets [26].

##### Adaptive Synthetic Sampling (ADASYN)

ADASYN shares the same algorithm for generating new samples as SMOTE; however, it also focuses on generating samples next to the originals, which are wrongly classified using a K-Nearest Neighbors (KNN) [27].

##### Borderline SMOTE

Borderline SMOTE is a variation of the SMOTE algorithm that supports the model that learns the dataset to train with difficult predictions by focusing on the samples that are near the borderline of the classes and generating those ‘hard samples’ [28].

#### 2.4.3. Learning Algorithms

Seven machine learning methods and two types of their ensembles were applied for predicting emergent CS during labor [22]. All of the algorithms were implemented based on the scikit-learn model application programming interface (API) (Logistic regression, Random Forest, Gradient boosting, Support vector machine (SVM), KNN, Voting, and Stacking), or compatible APIs supported by other libraries (light gradient boosting machine (LGBM) and extreme gradient boosting (XGBoost)) [29].

##### Logistic Regression

Logistic regression is a statistical technique that is often used to create a prediction model for categorical data. It predicts the probability value from 0 to 1 that a sample belongs to a specific category and classifies it when the probability value is greater than 0.5.

##### Random Forest

The random forests algorithm reduces overfitting by combining many weak learners that are under-fit using the subset of the dataset.

##### Gradient Boosting, LGBM, XGBoost

Gradient boosting, a boosting algorithm that starts with one node or a single leaf, has high performance, especially with tabular type data. The algorithm consists of many under-fitted models that sequentially decrease loss of the prediction. Three versions of Gradient boosting packages of scikit-learn, LGBM, and XGBoost were adopted in this study [30,31].

##### SVM

SVM creates and utilizes the boundary surface that separates samples in the dataset by class. By iteratively calculating and maximizing the margins between the surface and classes, SVM optimizes the position of the boundary surface for the data.

##### KNN

The KNN algorithm calculates the Euclidean distance between two random samples in the multi-dimension space and uses the nearest sample from the target data as reference for the prediction.

##### Ensembles: Voting, Stacking

Ensemble methods are techniques that create and combine multiple models to pro-duce improved results. Two ensemble algorithms of voting and stacking were used. With majority voting, the predicted class label for a particular sample is that representing the majority (mode) of the class labels predicted by each individual classifier. It would classify the sample as “class 1” based on the majority class label. The other algorithm, which was implemented based on stacking generalization, consists of stacking the output of the individual estimator and uses a classifier to compute the final prediction.

#### 2.4.4. Model Evaluation and Validation

Internal validation has been carried out using validation sets, and the most popular metrics were used to compare model performance. The accuracy score was adopted in order to evaluate the agreement between the prediction and true values. Precision and recall were also considered to measure how many of the predicted positives are truly positive and how many of the actual positives were correctly classified. In addition, the F1-score was calculated as the harmonic mean between precision and recall, as there is a tradeoff between those two measures.

Since it is necessary that the model outcome to be used be versatile, area under the receiver-operating characteristic curve (AUROC), which indicates the ability to separate positive classes from the negative class, was the first factor to consider; hence, it could represent overall model performance regardless of threshold. Moreover, there should be parameters defined as standards to discern the superior model among those that yield similar AUROC values. Simultaneously considering the other values allows for a comparison between the strengths and weaknesses, from which the most appropriate model could be discerned among multiple models. The recall score of the model had priority over accuracy in model evaluation depending on the algorithm used, as false negative classification leads to greater medical risk in the clinical environment.

#### 2.4.5. Software and Statistical Tools

We used the scikit-learn package for actual implementation of the machine learning algorithms and calculation of the metrics, including the scikit-learn-compatible APIs provided by LightGBM and XGBoost. In preprocessing imbalanced data, the imbalanced-learn package was used for the sampling algorithms. Pandas and Matplotlib were also used to appropriately process the tabular data as the training set for the model and to visualize the analysis plots [32,33]

### 2.5. External Validation

External validation was performed with the dataset of a multicenter retrospective cohort study of the Korean Society of Ultrasonography in Obstetrics and Gynecology (KSUOG), which was designed to analyze fetal growth percentile for the Korean population at 48 general hospitals in Republic of Korea and was approved by the institutional review board of each [34]. Ultrasonographic findings based on biometric findings from 34 weeks 0 days to 39 weeks 6 days’ gestation were collected. Women were aged 20–44 years, and gestational age at delivery varied from 24 weeks 0 days to 41 weeks 6 days in the original data. For external validation of our study, women who delivered a baby before 37 weeks 0 days or multiparous women were excluded. Women with elective CS (medically or surgically indicated CS or patient choice), unknown indication of CS, and CS before active labor (less than 4 cm dilatation of the cervical os with uterine contraction) were also excluded. The model performance of the external validation cohort was assessed for accuracy and C-statistics.

The Institutional Review Board of The Catholic University of Korea approved the present study (XC20WIDI0103/2020-2158-0020). Since this study was a retrospective cohort study and all data were anonymized, and the need for informed consent was waived.

## 3. Results

### 3.1. Patient Characteristics

The baseline characteristics of the study population (*n* = 6549) of 5507 (84.1%) women with vaginal delivery and 1042 (15.9%) women with emergent CS during labor are listed in Table 1. Maternal age, weight at pre-pregnancy and at the time of delivery, gestational age at delivery, neonatal weight at delivery, percentage of male babies, and PAH were higher in women with emergent CS during labor than in those who underwent vaginal delivery, while maternal height was significantly higher in women with vaginal delivery than in those with emergent CS. The BPD, AC, and EBW measured at 34 to 39 weeks of gestation were significantly higher in women with emergent CS than in those with vaginal delivery. There were no significant differences in gestational age at sonogram or gestational diabetes. Distributions of raw data are presented in Figure 2 and Figure 3.

### 3.2. Model Performance

The predictive performance of each algorithm is listed in Table 2. The calibration plot with the respective C-statistics of the machine learning models for predicting emergent CS are shown in Figure 4. The C-statistics for KNN, Voting, XGBoost, Stacking, gradient boosting, random forest, LGBM, logistic regression, and SVM were 0.6, 0.69, 0.64, 0.59, 0.66, 0.68, 0.68, 0.7, and 0.69, respectively. Based on the AUROC values, the logistic regression model had the best performance for predicting emergent CS. The overall accuracy of the logistic regression model was 0.78, specificity was 0.85, and sensitivity was 0.43.

### 3.3. Variable Influence on the Prediction Model

As the model based on logistic regression showed the best performance based on AUROC values, the nine significant variables associated with emergent CS during labor were identified by logistic regression analysis, as follows: (1) EBW (kg, adjusted odds ratio [aOR] (95% confidence interval [CI]), 1.13 (1.05–1.21)), (2) birth weight (kg, aOR (95% CI), 1.12 (1.09–1.16)), (3) PAH (aOR (95% CI), 1.09 (1.04–1.15)), (4) maternal age (year, aOR (95% CI), 1.01 (1.01–1.01)), (5) gestational age at delivery (kg, aOR (95% CI), 1.01 (1.001–1.021)), (6) maternal weight before pregnancy (kg, aOR (95% CI), 1.01 (1.004–1.006)), (7) maternal height (cm, aOR (95%CI), 0.99 (0.98–0.99)), (8) fetal sonographic AC (mm, aOR (95%CI), 0.98 (0.97–0.99)), and (9) fetal sonographic FL (mm, aOR (95% CI), 0.95 (0.91–0.99)) (Figure 5).

### 3.4. External Validation

Clinical and sonographic data of 1391 term nulliparous women were extracted in a fetal growth study cohort of the KSUOG and used for external validation of the logistic regression model trained from the original dataset. The results indicated that the C-statistics value for the model was 0.69, with an overall accuracy of 0.68, a specificity of 0.83, and a sensitivity of 0.41 (Figure 6). The confusion matrix of the prediction for an external validation set and available performance according to the threshold of the logistic regression model are presented in Figure 7 and Figure 8.

## 4. Discussion

In nine machine learning models, the logistic regression model had the best performance for predicting emergent CS. The significant variables were EBW, AC, and FL on ultrasonogram; neonatal birth weight; PAH; maternal age; gestational age at delivery; maternal weight at delivery; and maternal height. The overall accuracy of the logistic regression model was 0.78, with an area under the curve (AUC) of 0.70. In an external validation set, the overall accuracy of the logistic regression model was 0.68, with an AUC of 0.69.

Associations of parameters such as advanced maternal age, shorter status, and higher body mass index (BMI) with fetal sonographic parameters including large AC and emergent CS have been reported [11,12,35,36,37]. The findings of our study are consistent with those of previous studies [12,35,36,37]. We obtained additional information by integrating 12 variables into our prediction model. We observed that pregestational maternal weight was significantly associated with risk of emergent CS, while maternal weight at delivery did not increase the risk. Moreover, our prediction algorithm incorporated neonatal birth weight, gestational age at delivery, and major obstetric complications of GDM and PAH as variables. The risk of CS increases with increased neonatal birth weight [38]. Although the accuracy of US-based fetal weight estimation has improved in recent decades, estimated fetal weight remains inconsistent with actual birth weight [39,40]. Although data on neonatal birth weight are available after delivery, maternal motivation for controlling maternal and neonatal birth weights might affect maternal diet and exercise. Furthermore, considering the 39-week labor induction in nulliparous women, this result can be a reference for maternal counseling [41].

Some studies have reported that the size of the fetal head or abdomen alone is more strongly correlated with actual birth weight than is estimated fetal weight [21,37]. Others have suggested that fetal growth should be assessed using a separate biometric measure in addition to the estimated fetal weight to avoid a minimalist approach involving only a single value [39]. In our machine learning model, fetal EFW, BPD, FL, and AC were incorporated. Sonographic fetal head circumference (HC) ≥ 35 cm or 95th percentile, measured within 1 week of delivery, has shown significant association with emergency CS or instrumental delivery [33]. Therefore, it would be valuable to test whether adding a value of sonographic fetal HC into the machine learning model can improve performance when using a prospective design.

Our study has some limitations. This study was a retrospective cohort study. However, we tried to include all consecutive mothers who delivered during the study period to reduce selection bias. The obstetrician and patients were not blinded to the sonographic information obtained via fetal biometry. The estimation itself or knowledge of fetal weight might influence the rate of CS, regardless of actual birth weight [42]. This information might have influenced the obstetrician’s decision regarding mode of delivery and timing of labor induction. In addition, there are other factors such as cervical length, effacement, and fetal station that influence the success of vaginal delivery. Recent studies have started to use real-time data acquired throughout labor or preinduction parameters such as Bishop score, angle of progression, or cervical length by transvaginal ultrasonography to increase prediction accuracy for emergent CS [43,44]. Although this study did not include those intrapartum or preinduction data, our objective was prediction of emergent CS during labor in near term or term nulliparous women, because the maternal decision regarding the delivery mode is sometimes made before term or labor onset during the term. In addition, since this study was conducted only for Koreans, validation for other races is required for more widespread application [12,44]. However, considering that other prediction models mainly target western populations, and the predictive performance for Asians is low, this study is expected to be useful for East Asians.

One previous systematic review study revealed that the risk of developing depressive and posttraumatic stress disorder symptoms in women who requested elective CS but delivered vaginally was significantly higher than in women with normal vaginal delivery [19]. American College of Obstetricians and Gynecologist recommends vaginal delivery if there is no maternal or fetal indication for CS [45]. However, if a patient decides to pursue cesarean delivery on maternal request, cesarean delivery can be planned from 39 weeks’ gestation, after providing information about the increasing risks of placenta previa, placenta accrete spectrum, and gravid hysterectomy with each subsequent cesarean delivery [45]. In women requesting elective CS, fear of childbirth or emergency cesarean section in labor after a day of pain is one of the reasons for elective CS [46,47]. Although it is unclear whether knowledge of individual probability of emergent CS during labor or before the onset of labor can decrease the fear of childbirth, machine learning algorithms might provide more objective probability than an obstetrician’s subjective opinion.

One of the main strengths of this study is its multicenter design, which includes a heterogenous population from various hospital settings. The second strength of this study is the external validation using nationwide data from the KSUOG fetal growth cohort. Most prediction models in the area of obstetrics lack external validation [14,15,16,48,49]. Because the performance and impact of prediction models need to be assessed in clinical practice, the results should be validated externally in other medical settings and different populations. Because the performance of the logistic regression model was similar to that of the external validation, the model should be applicable in East Asian women. The third strength of this study is the use of machine learning methods, which demonstrated favorable prediction of emergency CS due to failure to progress or non-reassuring FHR. In addition, the probability of emergent CS can change according to maternal weight, gestational age at delivery, and neonatal birth weight. Last, this study incorporated parameters of common obstetric complications (e.g., GDM and PAH) for predicting emergency CS during labor.

## 5. Conclusions

Machine learning algorithms using maternal and ultrasonographic parameters obtained during pregnancy could effectively predict emergent CS during active labor. Although this prediction model does not include intrapartum parameters, most women want to know their individual probability of emergent CS during labor prior to the onset of labor. The findings obtained from our study are thought to be useful in providing contemporary antenatal care in nulliparous mothers with a greater fear of labor. This can provide individualized antenatal counseling of the risk of emergent CS, which can better inform women preparing for childbirth. In addition, this study can provide a quantified motivation for pregestational maternal weight management, which is a controllable factor in antenatal management, and can serve as a basis for discussing induced labor at an appropriate time.

If validated in other populations, the prediction algorithm could be useful for individualized counseling of nulliparous women. Further prospective studies including intrapartum clinical and ultrasonographic parameters are needed to improve prediction performance.

## Figures and Tables

**Figure 1 life-12-00604-f001:**
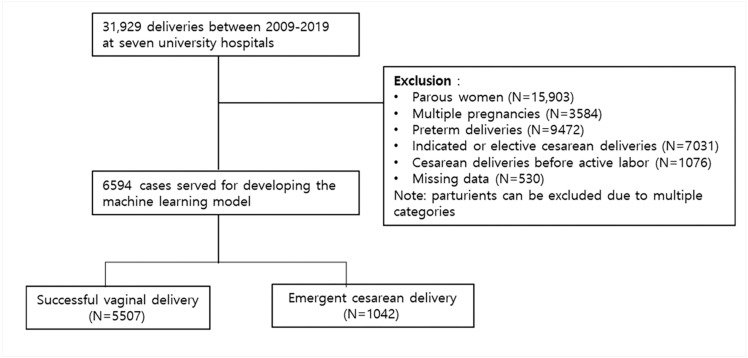
Participant selection process.

**Figure 2 life-12-00604-f002:**
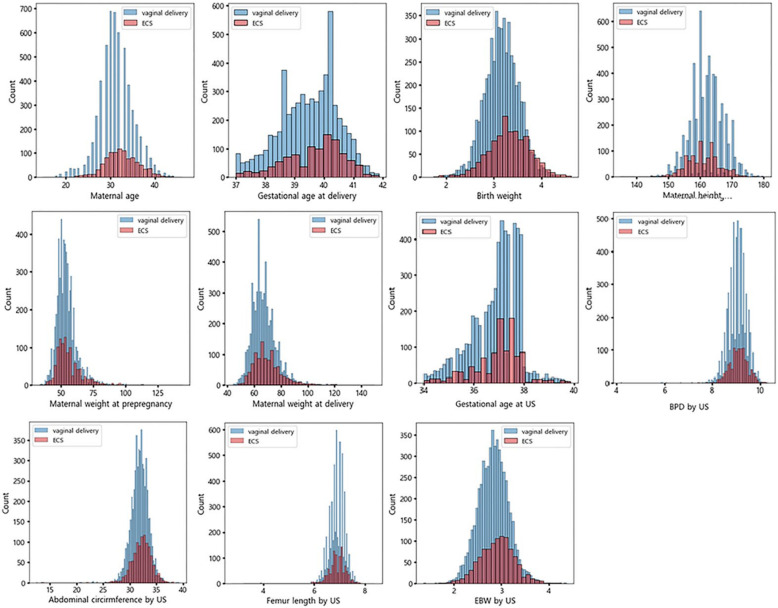
All-against-all scatter plot analysis of variables.

**Figure 3 life-12-00604-f003:**
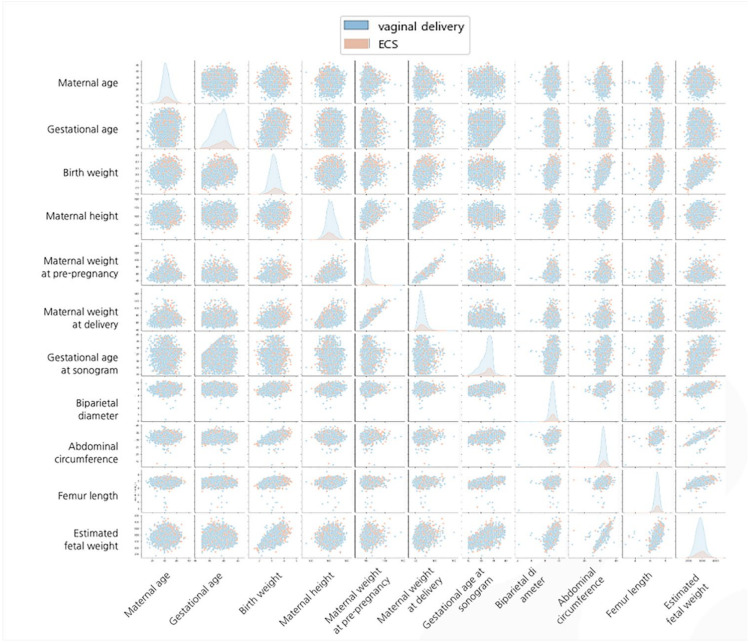
Correlations between delivery type and patient variables.

**Figure 4 life-12-00604-f004:**
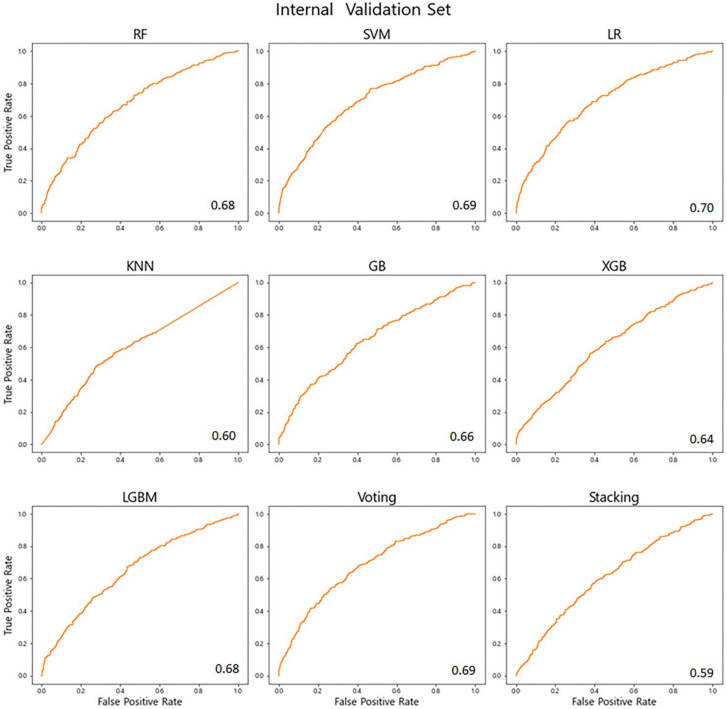
Receiver operating characteristic curves of emergency cesarean section prediction models.

**Figure 5 life-12-00604-f005:**
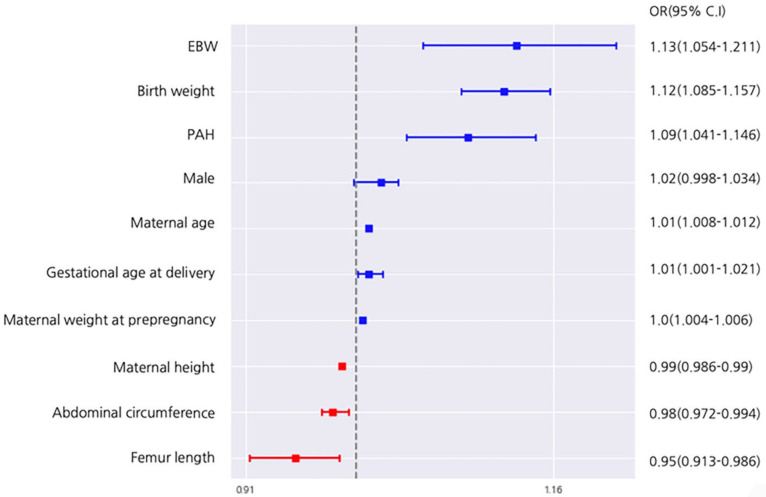
Odd ratios by logistic regression analysis.

**Figure 6 life-12-00604-f006:**
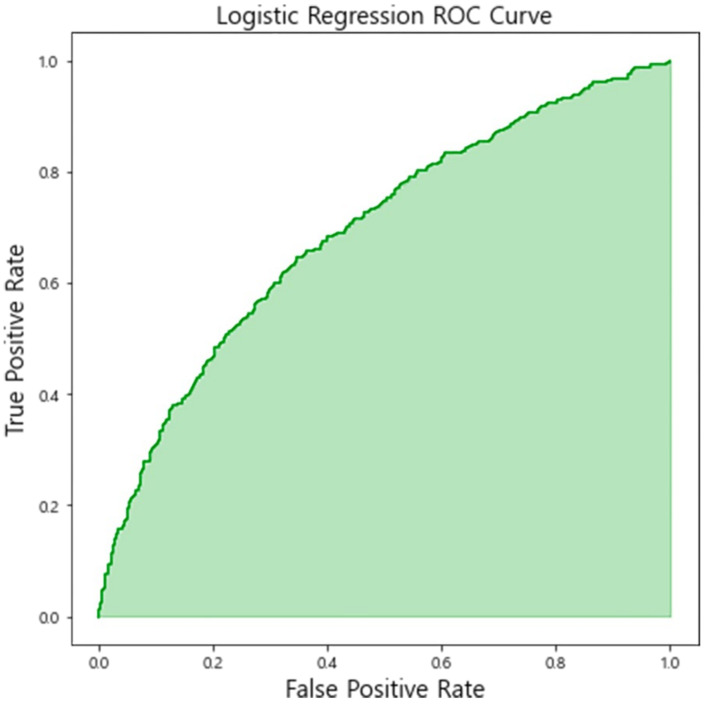
Receiver operating characteristic curve of emergency cesarean section prediction model based on logistic regression in an external validation set.

**Figure 7 life-12-00604-f007:**
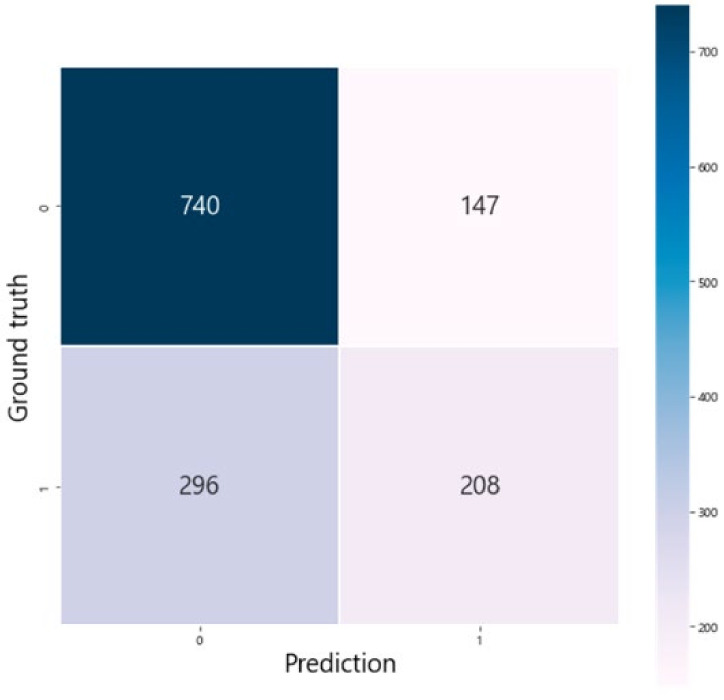
The confusion matrix of the prediction for an external validation set.

**Figure 8 life-12-00604-f008:**
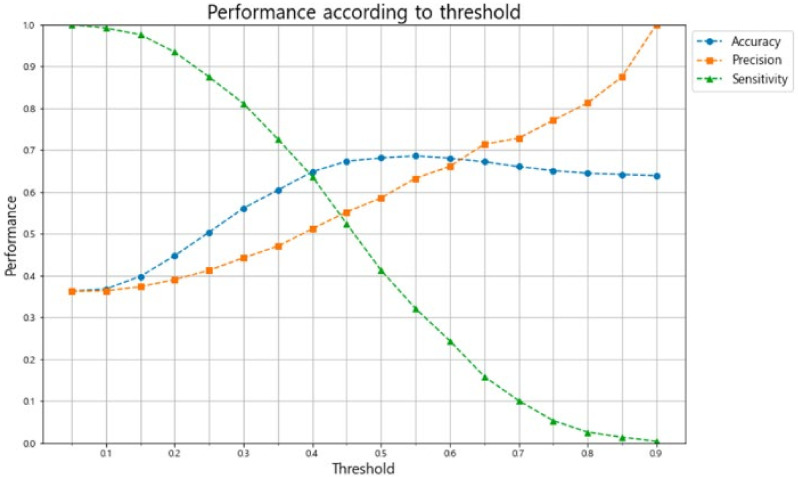
Model performance according to the threshold of the logistic regression model.

**Table 1 life-12-00604-t001:** Baseline characteristics of the study population.

Characteristics	Successful Vaginal Delivery (*n* = 5507)	Emergent Cesarean Section (*n* = 1042)	*p*-Value
Maternal age, yr	31.29 ± 3.8	32.49 ± 3.95	<0.001
Gestational age, w	39.49 ± 0.99	39.67 ± 0.99	<0.001
Birth weight, gm	3.17 ± 0.36	3.31 ± 0.44	<0.001
Maternal height, cm	162.24 ± 5.02	160.93 ± 5.14	<0.001
Maternal weight at pre-pregnancy, kg	53.96 ± 7.93	56.05 ± 9.97	<0.001
Maternal weight at delivery, kg	67.13 ± 9.11	69.92 ± 10.72	<0.001
Gestational age at sonogram, w	36.9 ± 0.95	36.96 ± 0.95	0.075
Sonographic parameters, cm			
Biparietal diameter	9.05 ± 0.37	9.09 ± 0.37	<0.001
Abdominal circumference	31.88 ± 1.62	32.23 ± 1.88	<0.001
Femur length	6.9 ± 0.31	6.92 ± 0.33	0.125
Estimated fetal weight	2831.84 ± 320.48	2908.89 ± 365.84	<0.001
Neonate male sex	2779 (50.46%)	580 (55.66%)	0.002
Pregnancy-associated hypertension	167 (3.03%)	61 (5.85%)	<0.001
Gestational diabetes	253 (4.59%)	62 (5.95%)	0.072

**Table 2 life-12-00604-t002:** Comparison of predictive performance for emergent cesarean section.

Algorithm	Accuracy	Precision	Sensitivity	F1_Score	Specificity
Logistic Regression	0.78	0.35	0.43	0.39	0.85
Voting	0.83	0.38	0.17	0.23	0.95
SVM	0.77	0.31	0.37	0.34	0.85
Random Forest	0.83	0.42	0.19	0.26	0.95
LGBM	0.85	0.55	0.15	0.23	0.98
Gradient Boosting	0.83	0.36	0.12	0.19	0.96
XGBoost	0.82	0.34	0.12	0.17	0.96
KNN	0.69	0.24	0.42	0.3	0.74
Stacking	0.83	0.35	0.09	0.14	0.97

SVM, Support Vector Machine; LGBM, Light Gradient Boosting Machine; KNN, k-nearest neighbors.

## Data Availability

Datasets used and/or analyzed during the current study are available from the corresponding author upon reasonable request.

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
