# Peer review of "Prediction of Emergency Cesarean Section Using Machine Learning Methods: Development and External Validation of a Nationwide Multicenter Dataset in Republic of Korea"

_life, 2022, doi:10.3390/life12040604_

Round 1
Reviewer 1 Report
Please find my comments in the attached file.

Author Response
Line 7: “Korean Society of Ultrasound in Obstetrics and Gynecology Research Group” this should be removed from authors list.
Answer) We corrected that as your comment.
Abstract: although it nicely written, I suggest that the last sentence should be revised and improved.
Answer) We revised the last sentence.
Introduction chapter needs significant improvement. Some of the paragraphs are poorly connected and objective of this study is poorly described. It would be nice if authors can introduce some information related to complications related to CS and pregnancy in general. How will this prediction help improve maternal and child health? Authors need to find a gap in our understanding on this title and then introduce what can be done and what they have studied in the present research.
Answer) We revised the introduction section.
In methods: Please clearly indicate why informed consent was waived off by the institutional board? This is comment is also for the publisher to check if this adheres to their policy.
Answer) Since this study was a retrospective cohort study and all data is anonymized, informed consent was waived by the institutional review boards. This comment added in the methods section.
In addition, I suggest authors to spilt the methods section into a few sub-sections with an aim of providing clarity regarding statistical analysis. In its current form it is very complex for general readership.
Answer) We have revised the method section of the paper to improve readability and clearly convey the meaning according to your comment.
Results: “3.1. Patient characteristics” lines 164-70 should be placed in the methods section. This is more of a methodologic and material data. The same applies to figure 1.
Answer) We have revised the text according to your comment.
Discussion is also biologically less relevant and should be optimized to cover the very objective of this study model. Other populations should also be kept in consideration.
Answer) As our study is a study for Koreans only, there are limitations in its application to the general population. This is further mentioned in the discussion section.
In conclusion: Lines 298-99-“Further prospective studies including intrapartum clinical and ultrasonographic parameters might be needed for improving the prediction performance”. This sentence should be optimized. Also remove “might be”.
Answer) We optimized the sentence as your recommendation.
English language and syntax need to be optimized.
Answer) We additionally performed English correction to optimize English language and syntax.

Reviewer 2 Report
In general, the study presented is extremely interesting and useful. However, I would like to make a few comments to the authors:
- Define CS abbreviations when entering for the first time (line 39).
- Lines 159-161: Unify with the content of lines 95-96.
- Lines 131-137: Introduce a brief explanation of the methods mentioned at some point in the research and reference each one for further consultation.
- Lines 143-145: Detail the use of the libraries justifying and detailing how they are applied in the proposed research.
- Lines 139-141: Define and comment on the named parameters and justify their use.
- Figure 2: Including the value of the C-statistics parameters in the corresponding curves.
- Figure 2: Varying the position before line 188 or after 197.
- Lines 202-210: Which of the proposed algorithms has been used in the use of backward elimination?
- Line 186 and 218: The materials appearing in supplements 1, 2, 3 and 4 are interesting and of sufficient importance to be included in the main document after the corresponding lines.
- Section 5: Conclusions should be lengthened. In my opinion, part of Section 4 should be moved to Section 5.
- Lines 246-260: It would be of great interest to develop, study and test the hypotheses introduced in this paragraph. Could the authors include a short discussion of how they believe such a study could be conducted in future research?
- Lines 281-286: Could the results obtained in this research be commented and compared with other works?
Author Response
Response to Reviewer 2 Comments
Define CS abbreviations when entering for the first time (line 39).
Answer) We have corrected this paper as your comment.
Lines 159-161: Unify with the content of lines 95-96.
Answer) We have corrected this paper as your comment.
Lines 131-137: Introduce a brief explanation of the methods mentioned at some point in the research and reference each one for further consultation.
Answer) We have corrected this paper as your comment.
Lines 143-145: Detail the use of the libraries justifying and detailing how they are applied in the proposed research.
Answer) We have corrected this paper as your comment.
Lines 139-141: Define and comment on the named parameters and justify their use.
Answer) We have corrected this paper as your comment.
Figure 2: Including the value of the C-statistics parameters in the corresponding curves.
Answer) We have corrected this paper as your comment.
Figure 2: Varying the position before line 188 or after 197.
Answer) We have corrected this paper as your comment.
Lines 202-210: Which of the proposed algorithms has been used in the use of backward elimination?
Answer) We have corrected this paper as your comment.
Line 186 and 218: The materials appearing in supplements 1, 2, 3 and 4 are interesting and of sufficient importance to be included in the main document after the corresponding lines.
Answer) We have corrected this paper as your comment.
Section 5: Conclusions should be lengthened. In my opinion, part of Section 4 should be moved to Section 5.
Answer) We have corrected this paper as your comment.
Lines 246-260: It would be of great interest to develop, study and test the hypotheses introduced in this paragraph. Could the authors include a short discussion of how they believe such a study could be conducted in future research?
Answer) We have corrected this paper as your comment.
Lines 281-286: Could the results obtained in this research be commented and compared with other works?
Answer) We have corrected this paper as your comment.

Round 2
Reviewer 1 Report
Thanks for addressing these issues. Manuscript has been improved after revision.